mathematical finance/artificial intelligence/statistics

AI ethics, artificial intelligence, economics, extreme value theory, financial regulation

**Author for correspondence:**
Nicholas Beale
e-mail: nicholas.beale@sciteb.com

# An unethical optimization principle

Nicholas Beale[1], Heather Battey[2], Anthony C. Davison[3] and Robert S. MacKay[4]

[1]Sciteb Ltd, 23 Berkeley Square, London W1J 6HE, UK
[2]Department of Mathematics, Imperial College London, 180 Queen's Gate, London SW7 2AZ, UK
[3]Institute of Mathematics, Ecole Polytechnique Fédérale de Lausanne, Station 8, 1015 Lausanne, Switzerland
[4]Mathematics Institute, University of Warwick, Coventry CV4 7AL, UK

NB, 0000-0003-0586-0045; HB, 0000-0001-9387-4628;
ACD, 0000-0002-8537-6191; RSM, 0000-0003-4771-3692

If an artificial intelligence aims to maximize risk-adjusted return, then under mild conditions it is disproportionately likely to pick an unethical strategy unless the objective function allows sufficiently for this risk. Even if the proportion $\eta$ of available unethical strategies is small, the probability $p_U$ of picking an unethical strategy can become large; indeed, unless returns are fat-tailed $p_U$ tends to unity as the strategy space becomes large. We define an unethical odds ratio, $Y$ (capital upsilon), that allows us to calculate $p_U$ from $\eta$, and we derive a simple formula for the limit of $Y$ as the strategy space becomes large. We discuss the estimation of $Y$ and $p_U$ in finite cases and how to deal with infinite strategy spaces. We show how the principle can be used to help detect unethical strategies and to estimate $\eta$. Finally we sketch some policy implications of this work.

## 1. Introduction

Artificial intelligence (AI) is increasingly deployed in commercial situations. Consider for example using AI to set prices of insurance products to be sold to a particular customer. There are legitimate reasons for setting different prices for different people, but it may also be profitable to 'game' their psychology or willingness to shop around. The AI has a vast number of potential strategies to choose from, but some are unethical—by which we mean, from an economic point of view, that there is a risk that stakeholders will apply some penalty, such as fines or boycotts, if they subsequently understand that such a strategy has been used. Such penalties can be huge: although these happened too early for an AI to be involved, the penalties levied on banks for misconduct are currently estimated to be over USD276 billion (see appendix A). In an environment in which decisions are increasingly made without human intervention, there is therefore a strong incentive to know under what circumstances AI systems might adopt unethical

strategies. Society and governments are closely engaged in such issues. Principles for ethical use of AI have been adopted at national [1] and international [2] levels and the area of AI ethics is one of very considerable activity [3,4]. Recent work has proposed a framework for developing algorithms that avoid undesirable outcomes [5].

Ideally there would be no unethical strategies in the AI's strategy space. But the best that can be achieved may be to have only a small fraction $\eta$ of such strategies being unethical. Unfortunately this runs up against the unethical optimization principle, which we formulate as follows.

*If an AI aims to maximize risk-adjusted return, then under mild conditions it is disproportionately likely to pick an unethical strategy unless the objective function allows sufficiently for this risk.*

# 2. Problem formulation

The following is a deliberately oversimplified representation that emphasizes certain aspects and ignores others. Consider an AI that is searching a strategy space $\mathcal{S}$ for a strategy $s$ that maximizes the risk-adjusted return for its owners, i.e. the return modified to account for the risk undergone in generating it. For brevity we shall drop the term 'risk-adjusted' after this paragraph. The AI seeks its strategy by attempting to maximize an apparent risk-adjusted return function $A(s)$. However, unknown to the AI, certain strategies in $\mathcal{S}$ would be considered unethical by stakeholders, who in the future may impose a penalty for adopting them. Such penalties may include fines, reparations, compensation and boycotts: what they have in common from our point of view is that they have a positive risk-adjusted cost which we denote by $C(s)$. We shall call the subset of $\mathcal{S}$ for which $C(s) > 0$ 'unethical' or Red, and the complementary subset, for which $C(s) = 0$, 'ethical' or Green. Hence the true risk-adjusted return $T(s)$ due to adoption of strategy $s$ may be expressed as

$$T(s) = A(s) - C(s) + Q(s), \tag{2.1}$$

where the 'error' $Q(s)$ accounts for other differences between $T(s)$ and $A(s)$ even when $C(s) = 0$, due to imperfections in the algorithm's capacity to predict the future accurately.

For example, in early 2018 a UK national newspaper reported [6] that several motor insurance companies quoted appreciably higher premiums for a fictitious driver named 'Mohammed Smith' than for one named 'John Smith', when all other data entered were identical. In this case the strategy space $\mathcal{S}$ would contain mappings $s$ from the data available to an insurance company to its quotes, and $A(s)$ would represent the apparent return to the company from adopting a particular mapping $s$. The company's true return $T(s)$ would depend on various factors that cannot be known when $s$ is chosen, such as the behaviour of those drivers who ask for quotes. The cost $C(s)$ to an insurance company of adopting an unethical strategy $s$ could include the financial impact of reputational damage, regulatory actions, and, if sued for discrimination, legal costs and payouts. The set Red would include mappings that used names in a way that was discriminatory, for example by race or gender, as well as any other unethical strategies, and the set Green would contain all other strategies in $S$. The error $Q(s)$ would represent possible differences between the true return $T(s)$ and the apparent return minus the cost, which might arise even if the latter was zero, i.e. even if $s$ was ethical.

Each term in (2.1) is treated as a random variable, the randomness arising from variation in the data available to the AI when determining $A(s)$, and from future events and data on which the cost $C(s)$, the true return $T(s)$, and thus $Q(s)$, also depend. Probabilistic operations below apply to the composite of these sources of randomness, because of our focus on understanding the general ethical considerations arising from such computations.

Let $p_U = \mathrm{Pr}(s^* \in \mathrm{Red})$ denote the probability that the chosen strategy

$$s^* = \mathrm{argmax}_{s \in \mathcal{S}} A(s)$$

is unethical, and assume there is some measure on $\mathcal{S}$, so one could in principle compute the proportion $\eta$ of $\mathcal{S}$ that is red. The green strategies comprise the remaining proportion $1 - \eta$ of $\mathcal{S}$. Then we can define an unethical odds ratio, denoted by capital upsilon,

$$Y := \frac{p_U}{1 - p_U} \div \frac{\eta}{1 - \eta}, \tag{2.2}$$

which represents the increase in odds of choosing an unethical strategy by using the AI, relative to choosing a strategy at random. A value of $Y$ close to unity will not represent a significant increase in

risk due to use of the AI, whereas if $Y \gg 1$ then the AI acts as a significant unethical amplifier. If $\eta$ equals 0.05 (or 0.01), for example, then having $Y = 10$ gives $p_U \approx 0.35$ (or 0.09).

If $T - Q$ has the same distribution on the red and green regions and the expected returns are finite, then

$$E(A \mid \text{Red}) = E(T - Q \mid \text{Red}) + E(C \mid \text{Red})$$
$$> E(T - Q \mid \text{Green}) = E(A \mid \text{Green}),$$

(2.3)

where, in a departure from conventional notation, we have written $E(A \mid \text{Red})$ as shorthand for $E\{A(s)\}$ for $s \in$ Red, etc., and equation (2.3) holds for any $s \in$ Red and any $s' \in$ Green.

Moreover, under mild conditions on the correlation of $C$ and $T - Q$, the variation of $C$ in Red but not in Green implies that s.d.$(A \mid \text{Red}) >$ s.d.$(A \mid \text{Green})$, if these standard deviations are finite. Thus below we shall suppose that the expected return in Red is $\Delta$ larger than that in Green, and that the standard deviation in Red is a factor $1 + \gamma$ larger than that in Green, i.e.

$$E(A \mid \text{Red}) = E(A \mid \text{Green}) + \Delta, \quad \text{s.d.}(A \mid \text{Red}) = (1 + \gamma)\text{s.d.}(A \mid \text{Green}).$$

As we shall see, the trade-off between returns from ethical and unethical strategies will depend on $\eta$, $\Delta$ and $\gamma$ and on the tail of the distribution of returns.

# 3. Asymptotic strategy space

Let $F$ be the cumulative distribution function (CDF) for the payoffs $A(s)$ with respect to the assumed measure on the green part of an infinite strategy space $\mathcal{S}$. Making more precise our assumptions about the red part, we assume that the CDF therein is $F_R(x) = F\{(x - \Delta)/(1 + \gamma)\}$. Although $\Delta$ and $\gamma$ were described above in terms of the expectation and standard deviation of returns, the argument does not require these moments to exist; $\Delta$ and $\gamma$ quantify the location and scale increases for red returns relative to green returns even if the expected return is infinite.

Suppose that $S$ strategies are drawn at random from $\mathcal{S}$ with respect to its assumed measure, and let $m$ denote the number of them that are unethical and $n$ the number that are ethical. By the law of large numbers, with large probability $m/S$ will be close to $\eta$ and $n/S$ to $1 - \eta$. Let $M_R$ and $M_G$ respectively denote the maximum payoffs for the $m$ red and $n$ green strategies. Then we would like to approximate $\Pr(M_R > M_G)$, the probability that the best strategy found is red.

In many cases the maximum $M_n$ of a random sample of size $n$ from a distribution $F$ can be renormalized using sequences $\{a_n\} > 0$ and $\{b_n\} \subset \mathbb{R}$ in order that $(M_n - b_n)/a_n$ converges as $n \to \infty$ to a limiting random variable $X$ having a generalized extreme-value distribution. This distribution has a tail index parameter $\xi$ that controls the weight of its right-hand tail, with increasing $\xi$ corresponding to fatter tails; it includes the Gumbel distribution $\exp\{-\exp(-x)\}$ as a special case for $\xi = 0$. Following the discussion above, we can write $M_R = \Delta + (1 + \gamma)M_m$ and $M_G = M_n$, where $M_m$ and $M_n$ are respectively the maxima of $m$ and $n$ mutually independent variables from $F$, and we suppose that $(M_m - b_m)/a_m$ and $(M_n - b_n)/a_n$ converge to variables $X$ and $W$, which are independent and have the same generalized extreme-value distribution. In appendix A, we obtain general expressions for the limiting probability $p_U$ under mild conditions, and compute $p_U$ and the unethical odds ratio $Y$ for some special cases:

—if $F$ is Gaussian, then the limiting variables $X$ and $W$ are Gumbel, and $Y \to \infty$ if $\Delta$, $\gamma$ or both are positive;
—if $F$ is log-Gaussian or exponential, then the limiting variables $X$ and $W$ are Gumbel and $Y \to \infty$ if $\gamma > 0$;
—if $F$ is Pareto, i.e. $F(x) = 1 - x^{-\nu}$ for $x > 1$ and $\nu > 0$, then $X$ and $W$ have Fréchet distributions with tail indexes $\xi = 1/\nu$, and

$$\lim_{S \to \infty} p_U = \frac{\eta(1 + \gamma)^\nu}{1 - \eta + \eta(1 + \gamma)^\nu},$$

(3.1)

which yields

$$Y \to Y^* = (1 + \gamma)^\nu \quad \text{as } S \to \infty;$$

(3.2)

and
—if $F$ is Student $t$ with $\nu$ degrees of freedom, then the Pareto limit applies.

The significance of these results is that if a large number of strategies is tested at random, then unless the distribution of the returns is fat-tailed, as in the cases of the Pareto or $t$ distributions, a responsible regulator or owner should be extremely cautious about allowing AI systems to operate unsupervised

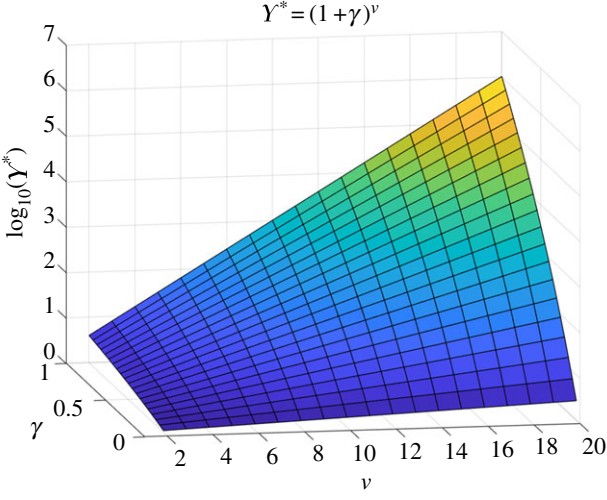

**Figure 1.** Dependence of the asymptotic unethical odds ratio $Y^*$ on tail index $\nu$ and additional volatility $\gamma$.

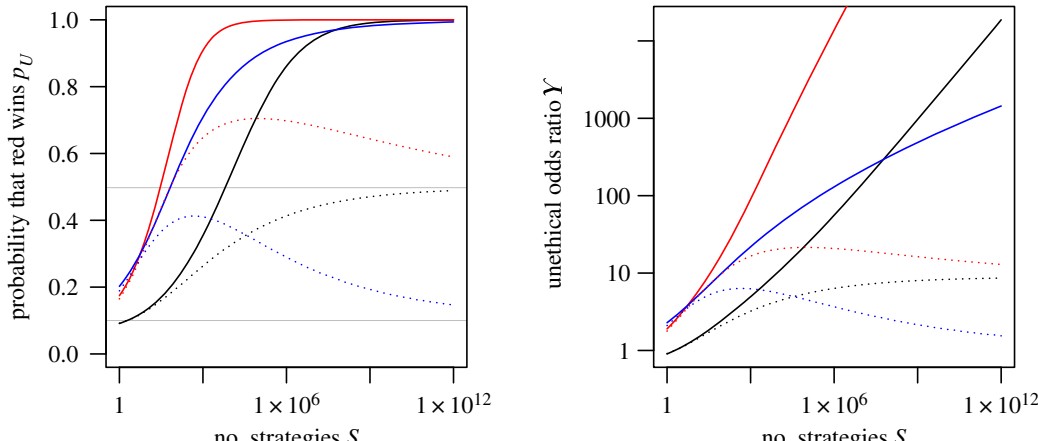

**Figure 2.** Dependence of probability $p_U$ and unethical odds ratio $Y$ on size of strategy space $\mathcal{S}$ for normal distribution (solid) and $t_{12}$ distribution (dots) when $\eta = 0.1$: $\gamma = 0.2$, $\Delta = 0$ (black); $\gamma = 0.2$, $\Delta = 0.5$ (red); $\gamma = 0$, $\Delta = 0.5$ (blue). The grey horizontal lines in the left-hand panel show the limiting probabilities from (3.1).

in situations with real consequences. If the returns are fat-tailed, then (3.2) gives some idea of the risk of using an unethical strategy.

Figure 1 shows how the tail index $\nu$ influences (3.2) in the heavy-tailed case. If $\nu = 7$, for example, then $Y^* \approx 1.4$ for $\gamma = 0.05$ and $Y^* \approx 17$ for $\gamma = 0.5$. For large $\gamma$ the value of $Y^*$ rises rapidly with $\nu$, and it remains small for all $\nu$ only when $\gamma \approx 0$.

# 4. Finite strategy space

For large but finite $S$ a simple and widely applicable algorithm to estimate $p_U$ and hence $Y$ is given in appendix A. Numerical experiments show that its limiting value $Y^*$ is reached quite rapidly for fat-tailed distributions, whereas $Y$ grows roughly as $\log S$ for Gaussian returns.

Figure 2 shows how the finite-sample unethical odds ratio $Y$ depends on $S$ for some special cases. In the Gaussian case the probabilities approach unity most rapidly when the volatility is inflated, i.e. $\gamma > 0$, and the unethical odds ratio appears to be ultimately log-linear in $\log S$. In the case of Student $t$ returns with $\nu = 12$ degrees of freedom, the probabilities overshoot their asymptotic values when $\Delta > 0$, and the asymptote (3.2) is approached rather slowly.

# 5. Correlated returns

So far we have represented the payoff function $A$ by independent draws from green or red distributions. A more general model is that $A$ is a random field over $\mathcal{S}$, dependence of which will result in nearby

strategies often having similar payoffs. Furthermore, the random field will not be stationary, in either its red or green regions; indeed, the concept of translations will not be meaningful. To obtain the probability that the maximum of $A$ is red requires specification of the random field and of the regions. One direction in which this can be addressed is presented in appendix A.

More generally, one could consider shades of red, corresponding to the likely size of the penalty and then ask for the distribution of redness for the maximum. Specifically, at every shade $r$ of red one could consider $p(r)\mathrm{d}r = \Pr\{s^* \in [r, r + \mathrm{d}r)\}$.

# 6. Estimating the parameters

The unethical optimization principle can help risk managers and regulators to detect unethical strategies. Consider a reasonably large sample $L \subset \mathcal{S}$. Manually examining $L$ for potential unethical elements may be prohibitively expensive if this requires human judgement. Suppose, however, that we rank the elements of $L$ by their values of $A(s)$ and focus our attention on the subset $L_k$ with the $k$ largest values of $A(s)$, where $k \ll |L|$. We assume that careful manual inspection can divide this set into red and green elements and write $\hat{p}_{U_k} = |L_k \cap \mathrm{Red}|/k$. By (2.2) we then have an estimator

$$\hat{\eta}_k = \frac{\hat{p}_{U_k}}{(1 - \hat{p}_{U_k})Y + \hat{p}_{U_k}}, \tag{6.1}$$

which provides a rough estimate of $\eta$ given $Y$ and $\hat{p}_{U_k}$. Perhaps more importantly, focusing on $L_k$ to find examples of unethical strategies that might be adopted not only weeds out those most likely to be used, but will help develop intuition on where problems might be found. Observing the bulk distribution of $A(s\,|\,s \in L)$ gives an idea of overall shape of $A(s)$ and an idea of $v$. To generate reasonably robust estimates of $\gamma$ and $\Delta$ it will generally be necessary to do some more manual inspection of another subset of $L$ to determine red and green elements but this can be relatively small if well targeted. Details are discussed in appendix A.

# 7. Implications

Ideally one would assign a measure of ethicality to strategies and adjust the objective function accordingly. If this is infeasible, practical advice to the regulators and owners of AI is to sample the strategy space and observe whether the returns $A(s)$ have a fat-tailed distribution. If not, then the 'optimal' strategies are likely to be unethical whatever the value of $\eta$. If, however, the observed return distribution is fat-tailed, then the tail index $v$ can be estimated using standard techniques [7,8] and $\eta$ can be estimated as discussed above. However, it would be unwise to place much faith in the precision of such estimates: there are so many imponderables that the main point is to avoid sailing close to the wind. In addition the principle can be used to help regulators, compliance staff and others to find problematic strategies that might be hidden in a large strategy space—the $k$ 'optimal' strategies can be expected to contain disproportionately many unethical ones, inspection of which should show where problems are likely to arise and thus suggest how the AI search algorithm should be modified to avoid them in future.

The principle also suggests that it may be necessary to re-think the way AI operates in very large strategy spaces, so that unethical outcomes are explicitly rejected in the optimization/learning process; see for example Thomas et al. [5] and Spiegelhalter [9].

This article introduces the unethical optimization principle and provides a simple formula to estimate its impact, as well as providing code for more detailed exploration. We hope that this quantitative connection between economics, financial regulation and AI ethics will provide a fruitful basis for discussion and for further research.

Data accessibility. This article has no additional data.

Authors' contributions. N.B. had the initial idea, formulated the principle, co-wrote the paper and performed some of the analytical work. H.B. indicated that the extremal types theorem could be used to quantify the risk in wide generality. R.S.M. did the initial analysis, leading to formulating the problem in terms of the odds ratio. A.C.D. provided most of the analysis and co-wrote the paper. All authors contributed importantly to the review, editing and revision of the paper, and did extensive background analysis.

Competing interests. At the time of writing, R.S.M. is an associate editor of Royal Society Open Science, but he had no involvement in the review or assessment of the paper.

Funding. This work was supported by the Swiss National Science Foundation, the UK Engineering and Physical Sciences Research Council grant number EP/P002757/1, Alan Turing Institute Fellowship TU/B/000101 and Capital International.

Acknowledgements. We thank Prof. He Ping, Deputy Governor Pan Gonsheng and Alex Brazier for organizing seminars at Tsinghua School of Economics and Management, the PBOC/SAFE and the Bank of England in March and April 2019 where N.B. presented the initial ideas that led to this paper. We also thank Andrew Bailey, Karen Croxson and Wolfram Peters for helpful discussions.

# Appendix A

## A.1. Recent penalties in financial services

The *Financial Times* listed [10] the major sets of fines and penalties levied on Western banks for various forms of misconduct. There were 11 types of misconduct and the fines and penalties totalled USD276 billion. Penalties (including compensation) for payment protection insurance totalled USD62 billion and was the second largest category.

## A.2. Derivation of limiting $p_U$

The extremal types theorem [11, theorem 1.4.2] implies that in wide generality, the maximum $M_n$ of a random sample $Z_1, \ldots, Z_n$ with cumulative distribution function $F$ may be renormalized using sequences $\{a_n\} > 0$ and $\{b_n\} \subset \mathbb{R}$ so that $(M_n - b_n)/a_n$ converges as $n \to \infty$ to a limiting random variable $X$ having a generalized extreme-value distribution. A simple sufficient condition for this is that $F(x)$ is twice continuously differentiable with probability density function $f(x)$ and that the derivative of the reciprocal hazard function $r(x) = \{1 - F(x)\}/f(x)$ converges to a constant $\xi$ as $x$ approaches the upper support point $x^*$ of $f$. Then we can take $b_n = F^{-1}(1 - 1/n)$, $a_n = r(b_n) > 0$ and the distribution of $X$ is

$$G_\xi(x) = \exp\{-(1 + \xi x)_+^{-1/\xi}\}, \quad x \in \mathbb{R}, \tag{A 1}$$

where $a_+ = \max(a, 0)$; setting $\xi = 0$ gives the Gumbel distribution $G_0(x) = \exp\{-\exp(-x)\}$. The quantity $\xi$, sometimes called the tail index, typically satisfies $|\xi| < 1$, with smaller values corresponding to lighter tails. If $\xi < 0$, then the limiting density has an upper support point at $-1/\xi$, whereas if $\xi \geq 0$ then the limiting density has no finite upper support point, so the limiting random variable has no upper bound.

This implies that we can write $M_n \approx b_n + a_n X$ for sufficiently large $n$, where the quality of the approximation depends on $F$; it has long been known that the convergence is extremely slow for Gaussian variables [12]. A result of Khintchine [11, theorem 1.2.3] implies that if $m = \eta S$ and $n = (1 - \eta)S$ for some fixed $\eta \in (0, 1)$, then as $S \to \infty$,

$$\frac{b_m - b_n}{a_n} \to \beta_\eta = \frac{\{\eta/(1 - \eta)\}^\xi - 1}{\xi}$$

and

$$\frac{a_m}{a_n} \to \alpha_\eta = \left(\frac{\eta}{1 - \eta}\right)^\xi,$$

with $\beta_\eta = \log\{\eta/(1 - \eta)\}$ when $\xi = 0$.

To apply these results, let $M_G$ denote the maximum of independent random variables $Z_1, \ldots, Z_n$ with common distribution function $F$, which represent the returns of ethical, green, strategies, and suppose that $(M_G - b_n)/a_n$ converges in distribution to a random variable $X$ as $n \to \infty$. Let $M_R$ denote the maximum of $m$ independent random variables $\Delta + (1 + \gamma)Z_j'$ representing the returns of unethical, red, strategies. We suppose that $Z_1', \ldots, Z_m'$ is a random sample from $F$ and that $\Delta \geq 0$ and $\gamma \geq 0$ quantify the increase in return and in volatility for unethical returns. We briefly discuss the case where the $Z_j$ and $Z_j'$ have different distributions below. Then

$$M_R = \Delta + (1 + \gamma)\max(Z_1', \ldots, Z_m'),$$

and as $S \to \infty$, $\{(M_R - \Delta)/(1 + \gamma) - b_m\}/a_m$ will converge in distribution to a random variable $W$ with the same distribution as $X$.

If $m$ is large enough, then we can write $M_R \approx \Delta + (1 + \gamma)b_m + a_m(1 + \gamma)W$, and so the probability that the best return from an unethical strategy exceeds the best return from an ethical one satisfies

$$\lim_{S \to \infty} \Pr(M_R > M_G) = \Pr\{\beta_\eta + A(\Delta, \gamma, \eta) + (1 + \gamma)\alpha_\eta W > X\},$$

where $A(\Delta, \gamma, \eta) = \lim_{S \to \infty} (\Delta + \gamma b_m)/a_n$ depends on $\eta$, $\Delta$, $\gamma$ and the normalizing sequences for maxima of random samples from $F$.

We now discuss the behaviour for large $S$ of

$$\frac{\Delta}{a_n} + \gamma \frac{b_m}{a_n} = \frac{\Delta}{a_n} + \gamma \frac{b_m}{a_m} \frac{a_m}{a_n}. \tag{A 2}$$

— If the upper support point $x^*$ is finite, then $a_n \to 0$ and $b_m/a_m \to \infty$, so $A(\Delta, \gamma, \eta) = \infty$. In this case the distributions of $M_G$ and $M_R$ become more and more concentrated for large $S$, and any advantage for Red leads to it beating Green with probability one, in the limit, because red returns have a higher upper limit than green ones.

— If the upper support point $x^*$ is infinite, then $a_n/b_n = r(b_n)/b_n \to \xi$ as $n \to \infty$, so $b_m/a_n = b_m/a_m \times a_m/a_n \to \xi^{-1}\alpha_\eta$, which is infinite if $\xi = 0$. The behaviour of $\Delta/a_n$ depends on the limit of $a_n = r(b_n)$ as $b_n \to \infty$. For example, if $F$ is exponential, then $a_n$ converges to a constant, whereas if $F$ is Gaussian, then $a_n \to 0$. For exponential maxima, therefore, $A(\Delta, \gamma, \eta)$ is infinite if $\gamma > 0$, but is finite if $\gamma = 0$, for any $\Delta$. For Gaussian maxima, $\xi = 0$ and $a_n \to 0$, so $A(\Delta, \gamma, \eta) = \infty$ if either of $\Delta$ or $\gamma$ is positive, i.e. if there is any systematic advantage for red strategies.

Other limits might appear when $\Delta$ and $\gamma$ depend on $S$, but one would need to consider whether this is realistic; for example, this might apply if $\eta \to 0$, i.e. red strategies are a vanishingly small fraction of all possible ones. This does not seem very realistic, since presumably any ethical strategy could be tweaked slightly to make it more profitable but unethical.

Here are the details for the special cases in the main text.

— If $F$ is Gaussian, then we can take $b_n = (2\log n)^{1/2}$ and $a_n = 1/b_n \to 0$, giving $\xi = 0$, so $\beta_\eta = \log\{\eta/(1 - \eta)\}$ and $\alpha_\eta = 1$. The limiting variables $X$ and $W$ are Gumbel, and Red will beat Green if either $\Delta$ or $\gamma$ is positive.

— If $F$ is log-Gaussian, then we can take $b_n = \exp\{(2\log n)^{1/2}\}$ and $a_n = b_n/(2\log n)^{1/2}$, so $\xi = 0$, $\beta_\eta = \log\{\eta/(1 - \eta)\}$ and $\alpha_\eta = 1$. The limiting variables $X$ and $W$ are Gumbel. Here $a_n \to \infty$ and $b_m/a_n \to \infty$, so Red always beats Green, owing to its higher volatility.

— If $F$ is exponential, then $b_n = \log n$, $a_n = 1$ and $\xi = 0$, so $X$ and $W$ are Gumbel, $\beta_\eta = \log\{\eta/(1 - \eta)\}$, $\alpha_\eta = 1$ and

$$\frac{(\Delta + \gamma b_m)}{a_n} = \Delta + \gamma \log S + \gamma \log \eta$$

tends to infinity unless $\gamma = 0$: Red beats Green in the limit owing to its higher volatility.

— If $F$ is Pareto, then $b_n = n^{1/\nu}$, $a_n = b_n/\nu$ and $\xi = 1/\nu$, so $\beta_\eta = \nu[\{\eta/(1 - \eta)\}^{1/\nu} - 1]$, $\alpha_\eta = \{\eta/(1 - \eta)\}^{1/\nu}$ and $A(\Delta, \gamma, \eta) = (1 + \gamma)\nu\alpha_\eta$. Here $X$ and $W$ have Fréchet distributions, $\exp\{-(1 + x/\nu)^{-\nu}\}$ for $x > -\nu$, and as $S \to \infty$, we obtain

$$\Pr(M_R > M_G) \to \frac{\eta(1 + \gamma)^\nu}{1 - \eta + \eta(1 + \gamma)^\nu}. \tag{A 3}$$

Hence $\Pr(M_R > M_G) > \eta$ for large $S$ if and only if $\gamma > 0$. This calculation also applies to other distributions with Pareto-like tails, such as the Student $t$. Inserting (A 3) into (2.2) yields (3.2).

The discussion above presupposes that the red and green returns only differ by a location and/or scale shift. If the limiting variables have the same support but different tail indexes, then the variable with the higher $\xi$ asymptotically dominates the other: if $W$ has a higher tail index than $X$, then red returns will beat green returns with probability one for large $S$.

## A.3. Computation of $p_U$

Let $m = S\eta$ and $n = S(1 - \eta)$. It is straightforward to check that

$$p_U = m \int F^n\{\Delta + (1 + \gamma)x\}f(x)F^{m-1}(x)\,dx,$$

which can be estimated by Monte Carlo simulation as follows:

—generate $U_1, \ldots, U_R \overset{\text{iid}}{\sim} U(0, 1)$, then set $M_r^* = F^{-1}(U_r^{1/m})$ for $r = 1, \ldots, R$;
—compute an estimate

$$p_1^* = R^{-1} \sum_{r=1}^{R} F\{\Delta + (1 + \gamma)M_r^*\}^n$$

of $p_U = \Pr(M_G \leq M_R)$;
—repeat the steps above, with $U_r^*$ replaced by $1 - U_r^*$ to give an estimate $p_2^*$;
—return $p_U^* = (p_1^* + p_2^*)/2$ as an estimate of $p_U$.

The first step uses inversion to generate maxima $M_r^*$ directly from $F^m$, the second step averages the exact probabilities $\Pr(M_G < M_r^*)$, and the third and fourth steps use antithetic sampling to reduce the variance of $p_U^*$. With $R = 10^5$ this gives probabilities accurate to three decimal places almost instantaneously. The R [13] code below embodies this.

```
prob.sim <- function(S, eta, delta, gamma, R=10^5)
{ # F is distribution function and Finv its inverse
 n <- (1-eta)*S
 m <- eta*S
 u <- runif(R)
 x <- Finv( u^(1/m) )
 m1 <- mean( F(delta+(1+gamma)*x)^n )
 x <- Finv( (1-u)^(1/m) )
 m2 <- mean( F(delta+(1+gamma)*x)^n )
 (m1+m2)/2
}
```

High-precision arithmetic may help in computing $p_U^*$ more accurately for very large $S$, though its precise value is rarely crucial.

Once $p_U$ has been estimated, $Y$ is obtained using equation (2.2).

## A.4. Correlated returns

As one example of the kind of approach discussed in the paper, consider the following.

Let $C(u, v)$ denote the copula that determines the dependence of random variables $U$ and $V$ having uniform marginal distributions. One standard measure of extremal dependence is [14]

$$\chi(u) = \Pr(U > u \mid V > u) = \frac{1 - 2u + C(u, u)}{1 - u}, \quad 0 < u < 1,$$

where $u \approx 1$ is of most interest in the present context. If $\chi = \lim_{u \to 1} \chi(u) > 0$, then $U$ and $V$ are said to be asymptotically dependent, with $\chi = 1$ corresponding to total dependence and $\chi = 0$ to so-called asymptotic independence. The quantity $2 - \chi$ can be roughly interpreted as the equivalent number of independent extremes at high levels of $(U, V)$, so $\chi = 1$ yields one 'equivalent independent' variable, and $\chi = 0$ yields two 'equivalent independent' variables. Rank-based estimators for $\chi(u)$ from independent data pairs $(u_1, v_1), \ldots, (u_n, v_n)$ are available for high values of $u$, e.g. $u = 0.95$. As these are based on the ranks, the marginal distributions of $U$ and $V$ are irrelevant.

To apply these ideas, suppose that $A(s)$ can be treated as a stationary process, that there is a measure of distance on $S$, and evaluate $A(s)$ on an equi-spaced grid, at $s \in 0, \pm \delta, \pm 2\delta, \ldots$, say. Thus we can observe the joint properties of $A(s)$ at distances $\delta, 2\delta, \ldots$, taking $U = A(s)$ and $V = A(s + k\delta)$ for each $s$ in the grid. If we take all such distinct pairs a distance $k\delta$ apart and estimate $\chi(0.95)$ as described above, then we can assess the dependence of the extremes of the process at lag $k$, for example by plotting the estimate $\hat{\chi}_k$

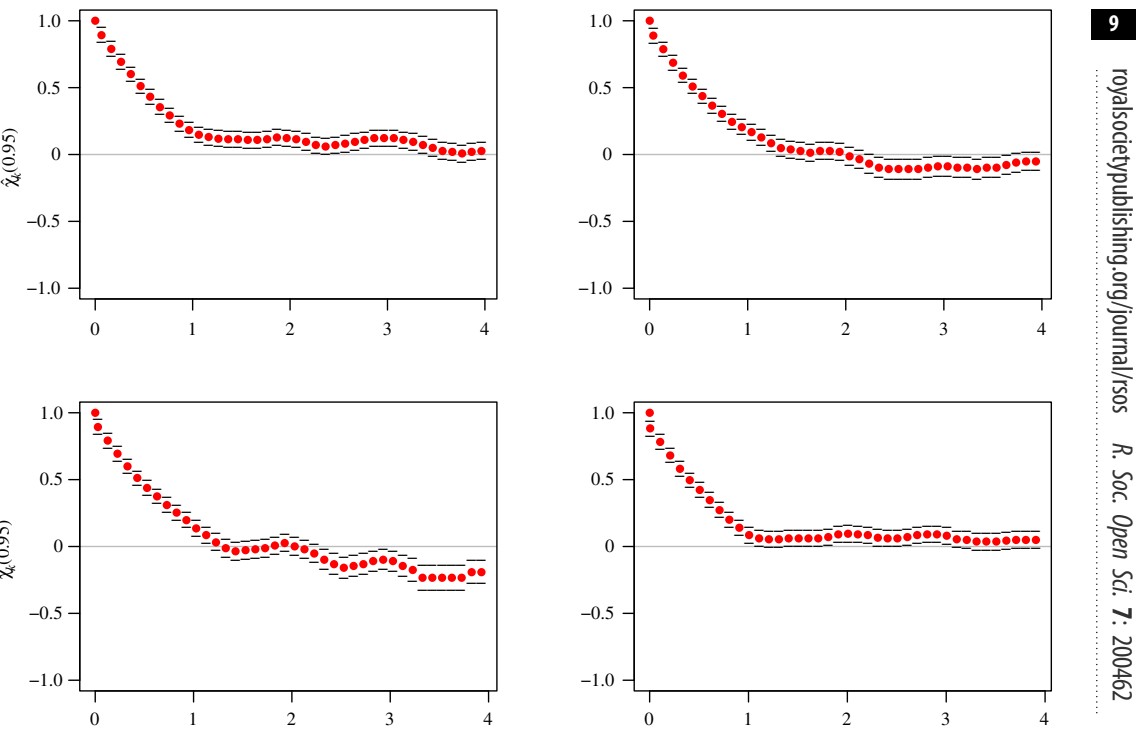

**Figure 3.** Four examples of $\hat{\chi}_k$ for the linear interpolation process described in the text. The red points show the estimates of $\chi(0.95)$ at different lags, and the tick marks show 95% confidence intervals for individual estimates. The sharp initial decline shows that local dependence of extrema of $A(s)$ becomes negligible when $k\delta > 1$ or so, as would be expected from the construction of $A(s)$.

against $k\delta$. This extremogram [15] will equal unity for $k = 0$, and should drop to zero as $k$ increases, and thus can be used to assess the approximate number of equivalent independent values in $\mathcal{S}$.

To illustrate this, we took $\mathcal{S} = [0, 1000]$, created a function $A(s)$ by linear interpolation between $S = 1001$ independent Gaussian variables at $s = 0, 1, \ldots, 1000$, and evaluated $A(s)$ on a grid with random initial value and $\delta = 0.1$. Figure 3 shows these plots for four simulated functions. The sampling properties of $\hat{\chi}_k$ for $k$ large mimic those for the usual time-series correlogram in the presence of strong dependence and are not good, but the sharp decline near the origin shows precisely the behaviour we expect; it appears that extreme values of $A(s)$ would be independent of those for $A(s \pm 2)$ or perhaps $A(s \pm 1)$, as we would anticipate from its construction. Thus if we sampled $\mathcal{S}$ at sites no closer than two units apart, the corresponding values of $A(s)$ could be taken as independent at extreme levels.

Although further refinement is certainly feasible, the discussion above suggests that it should be possible to identify an approximate number of 'independent' extrema in an infinite strategy space, under assumptions similar to those above, perhaps using a development of the ideas in Leadbetter [16].

## A.5. Estimation

To estimate the distributions for the ethical and unethical strategies, we suppose that the $k$ sampled strategies with the highest returns have been divided into $k_R$ unethical and $k_G$ ethical strategies, with respective returns $r_1, \ldots, r_{k_R}$ and $g_1, \ldots, g_{k_G}$, and we denote by $u$ the largest sampled return that is not among these $k$. In our asymptotic framework the generalized Pareto distribution (GPD) [17] provides a suitable probability model for $r_j - u$ and $g_j - u$, i.e. the 'excess' returns over $u$. The probability density functions for the red and green excesses are

$$\frac{1}{\tau_R}\left(1 + \xi\frac{r_j - u}{\tau_R}\right)_+^{-1/\xi-1}, \qquad \frac{1}{\tau_G}\left(1 + \xi\frac{g_i - u}{\tau_G}\right)_+^{-1/\xi-1},$$

for $j = 1, \ldots, k_R$ and $i = 1, \ldots, k_G$. The shape parameter $\xi$ is the same as in (A 1), and $\tau_R, \tau_G > 0$ are scale parameters. The effect of changes in both $\Delta$ and $\gamma$ appears in the ratio $\tau_R/\tau_G$, which will be larger than

**Table 1.** Summary results from simulation study with $\eta = 0.1$. $p_U$, $p'_U$ and $\hat{p}_U$, shown as percentages, are respectively the probability that red beats green, the average estimate of $p_U$ based on the top $k$ values, and the average estimate based on fitting generalized Pareto distributions to the red and green values. Power (%) is the estimated power for detecting a difference between the red and green samples. See text for details.

| distribution | $\Delta$ | $\gamma$ | $p_U$ | $p'_U$ | $\hat{p}_U$ | power |
|---|---|---|---|---|---|---|
| normal | 0 | 0 | 10.2 | 10.0 | 13.4 | 5.9 |
| | 0.5 | 0 | 41.4 | 25.7 | 47.7 | 19.3 |
| | 0 | 0.2 | 54.0 | 20.0 | 57.5 | 46.4 |
| | 0.5 | 0.2 | 86.8 | 38.6 | 90.3 | 90.4 |
| $t_{12}$ | 0 | 0 | 9.8 | 10.0 | 12.8 | 5.2 |
| | 0.5 | 0 | 20.4 | 21.3 | 25.4 | 5.4 |
| | 0 | 0.2 | 33.7 | 18.3 | 37.6 | 20.1 |
| | 0.5 | 0.2 | 50.1 | 32.1 | 58.4 | 33.0 |
| $t_4$ | 0 | 0 | 9.7 | 10.0 | 12.0 | 5.6 |
| | 0.5 | 0 | 11.8 | 15.6 | 13.6 | 6.8 |
| | 0 | 0.2 | 17.9 | 15.7 | 21.0 | 6.8 |
| | 0.5 | 0.2 | 20.7 | 22.8 | 24.3 | 5.1 |

unity if there is an advantage for red returns, whereas $\xi$ should be the same for red and green subsets. This last property is helpful: $\xi$ can be hard to estimate from small samples, but inference for it will be based on all $k$ of the largest returns. The adequacy of the GPD is readily checked using standard techniques [7, Ch. 4], and the parameters can be estimated, and models compared, using standard likelihood methods [18, Ch. 4].

Having obtained estimates $\hat{\xi}$, $\hat{\tau}_R$ and $\hat{\tau}_G$, we estimate $p_U$ by Monte Carlo simulation as follows. We generate standard uniform variables $U_1^*, \ldots, U_R^*$ and Poisson variables $N_1^*, \ldots, N_R^*$ with mean $r_{k_R}$, all mutually independent. We then compute $M_r^* = \hat{\tau}_R[\{1 - (U_r^*)^{1/N_r^*}\}^{-\hat{\xi}} - 1]/\hat{\xi}$, for $r = 1, \ldots, R$, and estimate $p_U$ by

$$\hat{p}_U = R^{-1} \sum_{r=1}^{R} \exp[-r_g\{1 - \hat{F}_G(M_r^*)\}],$$

where $\hat{F}_G$ denotes the fitted cumulative distribution function for the green exceedances over $u$, which is generalized Pareto with parameters $\hat{\xi}$ and $\hat{\tau}_G$. In the simulations described below we took $R = 10^5$, which reduces variation in $\hat{p}_U$ to the third decimal place.

We performed a small simulation experiment to check these ideas. For different settings with normal, $t_{12}$ and $t_4$ returns, we simulated 10 000 samples, each with $S = 10^4$ and $\eta = 0.1$. We constructed each sample by generating $Z_1, \ldots, Z_S \overset{\text{iid}}{\sim} F$, and then made red returns $\Delta + (1 + \gamma)Z_1, \ldots, \Delta + (1 + \gamma)Z_{S\eta}$, with the green returns being $Z_{S\eta+1}\ldots, Z_S$. We took the $k = 200$ largest returns for each sample, ascertained whether they were red or green, and obtained $u$, $r_1 - u, \ldots, r_{k_R} - u$ and $g_1 - u, \ldots, g_{k_G} - u$. We then fitted the GPD to the entire sample of $k$ excesses, and to the red and green excesses separately, using a common value of $\xi$; this enabled us to compute the likelihood ratio statistic for testing whether $\tau_R = \tau_G$, based on the $k$ largest returns; the proportion of times this is rejected is the statistical power for testing the hypothesis $\tau_R = \tau_G$ at a nominal 5% significance level. If the return distributions differ greatly, then this power should be high. We also computed the empirical value of $p_U$, based on whether the largest return in each sample was red or green, which would not be useful in practice, as it would equal either 0 or 1, based on the single sample available. As estimates of $p_U$ we computed the empirical proportion $p'_U = k_R/k$ and the estimate $\hat{p}_U$ described above, both of which would be available in practice.

Table 1 summarizes the results of this experiment. The rows with $\Delta = \gamma = 0$ show that when there is no difference between red and green returns $p_U$ and $p'_U$ are both close to the expected value of 10%, and the power is close to the anticipated value, 5%. Although $p'_U$ increases when either $\Delta$ or $\gamma$ is positive, in the normal and $t_{12}$ cases it generally has a downward bias and $\hat{p}_U$ appears to provide a better estimate of $p_U$. Computations not shown indicate that $\hat{p}_U$ can be highly variable, though taking $k = 500$ reduces its variance. The power increases in the normal case when $\Delta$ or $\gamma$ is positive, as predicted by the

asymptotic theory; when $\Delta = 0.5$ and $\gamma = 0.2$, for example, a difference between red and green returns can be detected in around 90% of samples. For the $t_{12}$ returns, $p_U$ and its estimates again increase, but more modestly, and more for increased volatility, $\gamma > 0$, than for increased mean, $\Delta > 0$. Again, this corresponds to the asymptotic theory. In the $t_4$ case neither $p'_U$ nor $\hat{p}_U$ dominates the other, and the power for detecting differences between red and green returns is very small; when $\eta = 0.1$, $\gamma = 0.2$ and $\nu = 4$, the limiting expression (3.1) yields $p_U \approx 0.15$, and low power is to be expected.

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
