## [Reviewer comments · Royal Society Open Science]

Review History

RSOS-200462.R0 (Original submission)

Review form: Reviewer 1

Is the manuscript scientifically sound in its present form?

No

Are the interpretations and conclusions justified by the results?

Yes

Is the language acceptable?

Yes

Do you have any ethical concerns with this paper?

No

Have you any concerns about statistical analyses in this paper?

Yes

Recommendation?

Major revision is needed (please make suggestions in comments)

Comments to the Author(s)

See report (Appendix A).

Review form: Reviewer 2

Is the manuscript scientifically sound in its present form?

Yes

Are the interpretations and conclusions justified by the results?

Yes

Is the language acceptable?

Yes

Do you have any ethical concerns with this paper?

No

Have you any concerns about statistical analyses in this paper?

No

Recommendation?

Accept with minor revision (please list in comments)

Comments to the Author(s)

The paper is nicely written and discusses an interesting aspect regarding the application of AI for strategy selection. The authors find that, unless returns have a fat-tailed distribution, AI tends to select an unethical strategy. This result is supported by calculating the probability of choosing an unethical strategy for a range of probability distributions for the risk-adjusted returns. However, a few aspects should be clarified and some minor typos need to be corrected. Please find my detailed comments below:

1. p.3, line 34: "an" is missing before "apparent risk-adjusted return function A(s),...
2. p.6, line 32: Please clarify what you mean by "distribution of redness for the maximum".
3. p.6, line 42: The intersection should be of "L_k and Red" and not "L and Red"
4. p.6, line 51: "SI" is used instead of "Supporting Information", while this is not the case in the rest of the manuscript.
5. Page 7: It seems to me that there is a discrepancy between the statements about the Unethical Optimization Principle in the Introduction and the Implications. The principle in the Introduction focuses on adjusting the objective function in order to avoid unethical strategies, while the Implications state that it can be used to find problematic strategies.
6. page 7: Please provide a little more detail / discussion on how knowledge of the tail index may affect someone's approach to find a good ethical strategy.

Decision letter (RSOS-200462.R0)

19-May-2020

Dear Professor Davison

On behalf of the Editors, I am pleased to inform you that your Manuscript RSOS-200462 entitled "An Unethical Optimization Principle" has been accepted for publication in Royal Society Open Science subject to minor revision in accordance with the referee suggestions. Please find the referees' comments at the end of this email.

The reviewers and handling editors have recommended publication, but also suggest some minor revisions to your manuscript. Therefore, I invite you to respond to the comments and revise your manuscript.

- Ethics statement

- Data accessibility

If you wish to submit your supporting data or code to Dryad (<http://datadryad.org/>), or modify your current submission to dryad, please use the following link:
<http://datadryad.org/submit?journalID=RSOS&manu=RSOS-200462>

- Competing interests

- Authors' contributions

- Acknowledgements

- Funding statement

Please ensure you have prepared your revision in accordance with the guidance at <https://royalsociety.org/journals/authors/author-guidelines/> -- please note that we cannot publish your manuscript without the end statements. We have included a screenshot example of

the end statements for reference. If you feel that a given heading is not relevant to your paper, please nevertheless include the heading and explicitly state that it is not relevant to your work.

Because the schedule for publication is very tight, it is a condition of publication that you submit the revised version of your manuscript before 28-May-2020. Please note that the revision deadline will expire at 00.00am on this date. If you do not think you will be able to meet this date please let me know immediately.

If your manuscript is newly submitted and subsequently accepted for publication, you will be asked to pay the article processing charge, unless you request a waiver and this is approved by Royal Society Publishing. You can find out more about the charges at <https://royalsocietypublishing.org/rsos/charges>. Should you have any queries, please contact openscience@royalsociety.org.

on behalf of Dr Kevin Glazebrook (Associate Editor) and Mark Chaplain (Subject Editor)
openscience@royalsociety.org

Associate Editor Comments to Author (Dr Kevin Glazebrook):

Comments to the Author:

This is a very interesting and thought provoking paper in an area of great current interest. The reviewers have raised a range of issues, most of which I am hopeful that the authors can deal with readily. Can I strongly encourage the authors to respond to all comments of both reviewers?

Reviewer comments to Author:
Reviewer: 1

Comments to the Author(s)
See report.

Reviewer: 2

Comments to the Author(s)

The paper is nicely written and discusses an interesting aspect regarding the application of AI for strategy selection. The authors find that, unless returns have a fat-tailed distribution, AI tends to select an unethical strategy. This result is supported by calculating the probability of choosing an unethical strategy for a range of probability distributions for the risk-adjusted returns. However, a few aspects should be clarified and some minor typos need to be corrected. Please find my detailed comments below:

1. p.3, line 34: "an" is missing before "apparent risk-adjusted return function A(s),..."
2. p.6, line 32: Please clarify what you mean by "distribution of redness for the maximum".
3. p.6, line 42: The intersection should be of "L_k and Red" and not "L and Red"
4. p.6, line 51: "SI" is used instead of "Supporting Information", while this is not the case in the rest of the manuscript.
5. Page 7: It seems to me that there is a discrepancy between the statements about the Unethical Optimization Principle in the Introduction and the Implications. The principle in the Introduction focuses on adjusting the objective function in order to avoid unethical strategies, while the Implications state that it can be used to find problematic strategies.

6. page 7: Please provide a little more detail / discussion on how knowledge of the tail index may affect someone's approach to find a good ethical strategy.

Author's Response to Decision Letter for (RSOS-200462.R0)

See Appendix B.

Decision letter (RSOS-200462.R1)

05-Jun-2020

Dear Professor Davison,

It is a pleasure to accept your manuscript entitled "An Unethical Optimization Principle" in its current form for publication in Royal Society Open Science.

on behalf of Dr Kevin Glazebrook (Associate Editor) and Mark Chaplain (Subject Editor)
openscience@royalsociety.org

Appendix A

Referee Report on “An Unethical Optimization Principle”

May 10, 2020

1 Content

This article introduces an unethical optimisation principle for returns of an AI based decision and provides tools to analyse and estimate it. It is shown that unless the returns are fat-tailed, the probability of choosing an unethical strategy goes to one.

2 Comments

I believe that this article addresses important questions of fairness of AI based decision making. I understand that this is a journal that aims at reaching a broad audience and that mathematical details must be kept to a minimum (at least in the main text). It would however be helpful for expert readers to have more information on the precise setup in order to enable a clear judgment of the underlying model assumptions.

- Could you give a concrete example for an application and the functions A , C and Q , and the space \mathcal{S} . In particular, can you mention which of these function are random and where sources of randomness come from.
- What is $A(s)$ exactly? Is it a prediction of an AI algorithm on the return of strategy s (excluding the impact of potential penalties)? But then in what sense is $A(s)$ random? The prediction should be non-random whereas the true outcome should be random. Is not all randomness covered by $Q(s)$?
- For each strategy $s \in \mathcal{S}$, there seems to be a (random) return with distribution F . The company chooses the (random) optimal strategy

$$s^* = \operatorname{argmax}_{s \in \mathcal{S}} A(s).$$

If this is correct, maybe it would make sense to define s^* somewhere.

- It seems like $A(s)$ has the same distribution for all green strategies, and another distribution for all red strategies. Is that not overly simplistic since it means that basically there is no difference in terms of the returns whatever strategy you choose (only between green or red strategies).

- I have a hard time imagining a the setup of the paper in practice. When do I have a fixed set of strategies and a (known) probability measure on this set? What would that probability measure represent? Can you give a concrete example for this?
- Related to the above comment: in section 5 you talk about correlated returns as realizations of a random field. What is an example for such a setup?
- What do you mean by “risk-adjusted”?
- p4l6: A more direct formulation would be easier to read: “If $T - Q$ has the same distribution on Red and Green regions and the mean returns are finite”
- p4l13: Do you assume independence between T , Q and C ? If yes, write it somewhere.
- p4l16: Can you define Δ also in formulas. In particular, is this the expectation of A or of T . Same question for γ .
- p4l46: It seems like you must require that the variance of the distribution F exists in order to define γ . In your example with Pareto and Student t distribution that would mean that $\nu > 2$. This also concerns figure 1. It is a bit curious that this restriction cannot be seen in formula (3.1) as a singularity at 2; or am I missing something?
- p8l53: The equality in distribution is actually a usual equality =
- (e) Estimation: Why did you choose Student t with $\nu = 12$? This seems to be already fairly close to the Gaussian case. Would it not be more interesting to look at something with heavier tails to contrast the Gaussian case?

Appendix B

We thank all the reviewers for the time they devoted to our work, their positive reactions and constructive comments, and hope that they have been dealt with satisfactorily. In addition to the changes noted below, we have made a few minor clarifications to the text.

Our replies below are in red.

Associate Editor

This is a very interesting and thought provoking paper in an area of great current interest. The reviewers have raised a range of issues, most of which I am hopeful that the authors can deal with readily. Can I strongly encourage the authors to respond to all comments of both reviewers?

Thank you for your positive reaction. We hope you find our responses satisfactory.

Reviewer 1

Content

This article introduces an unethical optimisation principle for returns of an AI based decision and provides tools to analyse and estimate it. It is shown that unless the returns are fat-tailed, the probability of choosing an unethical strategy goes to one.

Comments

I believe that this article addresses important questions of fairness of AI based decision making. I understand that this is a journal that aims at reaching a broad audience and that mathematical details must be kept to a minimum (at least in the main text). It would however be helpful for expert readers to have more information on the precise setup in order to enable a clear judgment of the underlying model assumptions.

- Could you give a concrete example for an application and the functions A , C and Q , and the space \mathcal{S} ? In particular, can you mention which of these function are random and where sources of randomness come from.

We've added to the paper a briefer version of the text below:

For example, in early 2018 a UK national newspaper reported that several motor insurance companies quoted appreciably higher premiums for a fictitious driver named 'Mohammed Smith' than for one named 'John Smith', when all other data entered were identical. In this case the strategy space \mathcal{S} would contain mappings s from the data available to an insurance company to its quotes, and $A(s)$ would represent the apparent return to the company from adopting a particular mapping s . The company's true return $T(s)$ will depend on various factors that cannot be known when s is chosen, such as the behaviour of the customers that ask for quotes. The cost $C(s)$ to an insurance company of adopting an unethical strategy s could include the financial impact of reputational damage, regulatory actions, and, if sued for discrimination, legal costs and payouts. The set Red would include mappings that used names in a way that was discriminatory, for example by race or gender, as well as any other unethical strategies, and the set Green would then contain all

other strategies in S . The error $Q(s)$ would represent possible differences between the true return $T(s)$ and the apparent return minus the cost, which might arise even if the latter was zero, i.e., even if s was ethical.

Any mathematical formulation is a highly idealised model of a complex reality. In the present setting randomness is used to describe variation, which can arise through the data \mathcal{D} used to choose the strategy s , through future data \mathcal{F} to which s will be applied and through the algorithm used to choose s ; we shall ignore the last of these. Suppose that \mathcal{D} represents a sample from a larger population from which \mathcal{F} is also drawn, though \mathcal{F} and \mathcal{D} may have different characteristics owing to population drift, non-random sampling, etc. Then we can write (2.1) of the paper more explicitly as

$$T(s; \mathcal{D}, \mathcal{F}) = A(s; \mathcal{D}) - C(s; \mathcal{D}, \mathcal{F}) + Q(s; \mathcal{D}, \mathcal{F}).$$

From the viewpoint of the AI, \mathcal{D} represents fixed data, but \mathcal{F} is as yet unobserved and thus may be treated as random, whereas we take the viewpoint of a regulator attempting to set a general framework for the application of AIs and treat both \mathcal{D} and \mathcal{F} as random.

- What is $A(s)$ exactly? Is it a prediction of an AI algorithm on the return of strategy s (excluding the impact of potential penalties)? But then in what sense is $A(s)$ random? The prediction should be non-random whereas the true outcome should be random. Is not all randomness covered by $Q(s)$?

See our response to the previous point.

- For each strategy $s \in \mathcal{S}$, there seems to be a (random) return with distribution F . The company chooses the (random) optimal strategy $s^* = \operatorname{argmax}_{s \in \mathcal{S}} A(s)$. If this is correct, maybe it would make sense to define s^* somewhere.

Done.

- It seems like $A(s)$ has the same distribution for all green strategies, and another distribution for all red strategies. Is that not overly simplistic since it means that basically there is no difference in terms of the returns whatever strategy you choose (only between green or red strategies).

The distributions are idealisations that represent the ranges of returns that may arise due to adopting different strategies.

- I have a hard time imagining a the setup of the paper in practice. When do I have a fixed set of strategies and a (known) probability measure on this set? What would that probability measure represent? Can you give a concrete example for this?

For the types of application we have in mind, \mathcal{S} could be very complicated — for instance, it might represent a large set of weights and thresholds of the interconnects on each layer of a neural net, or the parametrization of a random forest or a Gaussian process. To avoid giving a detailed specification we only require that there is a measure μ on \mathcal{S} so that $\eta = \mu(\text{Red})/\mu(\mathcal{S})$. Thus, if \mathcal{S} is a compact subset of \mathbb{R}^d , μ could be Lebesgue measure, whereas if a Bayesian approach is used, μ would correspond to the prior measure on the parameter space.

- Related to the above comment: in section 5 you talk about correlated returns as realizations of a random field. What is an example for such a setup?

As above, write

$$T(s; \mathcal{D}, \mathcal{F}) = A(s; \mathcal{D}) - C(s; \mathcal{D}, \mathcal{F}) + Q(s; \mathcal{D}, \mathcal{F}).$$

Then if we regard \mathcal{D} as random, $A(s) = A(s; \mathcal{D})$ is a random field over \mathcal{S} with covariance function $C(s, t) = E[\{A(s) - E[A(s)]\}\{A(t) - E[A(t)]\}]$, where the expectation is taken over the distribution of \mathcal{D} . If \mathcal{S} is a metric space, then there is a measure of closeness of strategies s and t and it would be natural to suppose that the correlation between $A(s)$ and $A(t)$ depends on the distance between s and t , with similar strategies yielding similar apparent returns. This would be the case if, for example, \mathcal{S} was a subset of \mathbb{R}^d for some (probably very large) d .

- What do you mean by ‘risk-adjusted’?

In its simplest definition, risk-adjusted return measures how much return an investment has made relative to the amount of risk the investment has taken over a given period of time. If two or more investments have the same return over a given time period, the one with the lowest risk will have the better risk-adjusted return. Thus other things being equal a certain return of 10% is considered preferable to a 10% chance of a 100 return and a 90% chance of nothing, as the mean return is the same, but the second has a higher variance. In practice there are all kinds of complications; for example different measures of risk could be used and might disagree, but the basic insight holds. For an overview see, for example, *Principles of Corporate Finance*, Brealy, Myers & Allen (2019) McGraw-Hill (ISBN 978-1-260-56555-3). We’ve added a brief comment about this in the paper.

- p416: A more direct formulation would be easier to read: ‘If $T - Q$ has the same distribution on Red and Green regions and the mean returns are finite.’

Thanks. Done.

- p4113: Do you assume independence between T , Q and C ? If yes, write it somewhere.

The text has been rewritten to clarify the argument.

- p4116: Can you define Δ also in formulas. In particular, is this the expectation of A or of T . Same question for γ .

Done.

- p4146: It seems like you must require that the variance of the distribution F exists in order to define γ . In your example with Pareto and Student t distribution that would mean that $\nu > 2$. This also concerns figure 1. It is a bit curious that this restriction cannot be seen in formula (3.1) as a singularity at 2; or am I missing something?

Although Δ and γ are motivated in terms of mean and standard deviations, the mathematical argument treats them simply as changes in the location and scale of the distribution of returns, so the argument does not require F to have any moments;

that's why $\nu = 2$ has no special role in (3.1). A comment to this effect is now made in the text.

- p8l53: The equality in distribution is actually a usual equality =.

Thanks.

- (e) Estimation: Why did you choose Student t with $\nu = 12$? This seems to be already fairly close to the Gaussian case. Would it not be more interesting to look at something with heavier tails to contrast the Gaussian case?

$\nu = 12$ was taken for consistency with Figure 2. We've now added results for $\nu = 4$ to the table. In this case the limiting probability in (3.1) is 0.15, so the power for detecting a difference between red and green samples is very low; the estimators perform about as well as in the other cases.

Reviewer 2

The paper is nicely written and discusses an interesting aspect regarding the application of AI for strategy selection. The authors find that, unless returns have a fat-tailed distribution, AI tends to select an unethical strategy. This result is supported by calculating the probability of choosing an unethical strategy for a range of probability distributions for the risk-adjusted returns. However, a few aspects should be clarified and some minor typos need to be corrected. Please find my detailed comments below:

1. p.3, line 34: "an" is missing before "apparent risk-adjusted return function $A(s)$:"

Done.

2. p.6, line 32: Please clarify what you mean by "distribution of redness for the maximum":

Done.

3. p.6, line 42: The intersection should be of " L_k and Red" and not " L and Red":

Done.

4. p.6, line 51: "SI" is used instead of "Supporting Information", while this is not the case in the rest of the manuscript.:

As the paper is relatively short and there are no size limits for the journal, we've changed SI/Supporting Information/... to Appendix throughout, and left the material at the end of the paper, for the convenience of any interested readers.

5. Page 7: It seems to me that there is a discrepancy between the statements about the Unethical Optimization Principle in the Introduction and the Implications. The principle in the Introduction focuses on adjusting the objective function in order to avoid unethical strategies, while the Implications state that it can be used to find problematic strategies.

Thank you. We have slightly enlarged the first paragraph of the Implications section.

6. Page 7: Please provide a little more detail / discussion on how knowledge of the tail index may affect someone's approach to find a good ethical strategy.

It's not so much finding a good ethical strategy as avoiding bad ones. Having humans inspect the whole of \mathcal{S} is likely to be prohibitively expensive. But if they inspect the k 'optimal' strategies they are disproportionately likely to spot unethical ones (such as apparent discrimination based on ethnic names), and once a generic type of unethical strategy is identified, other instances should be found more readily.